# Exploring the Health-Related Quality of Life and the Lived Experience of Adolescents Following Invasive Meningococcal Disease

**DOI:** 10.3390/healthcare12111075

**Published:** 2024-05-24

**Authors:** Mark McMillan, Joshua McDonough, Margaret Angliss, Jim Buttery, Lynda Saunders, Suja M. Mathew, David Shaw, David Gordon, Morgyn S. Warner, Renjy Nelson, Rory Hannah, Helen S. Marshall

**Affiliations:** 1Vaccinology and Immunology Research Trials Unit, Women’s and Children’s Health Network, Adelaide, SA 5006, Australia; mark.mcmillan@adelaide.edu.au (M.M.); lynda.saunders@sa.gov.au (L.S.); suja.mathew@adelaide.edu.au (S.M.M.); 2Robinson Research Institute and Adelaide Medical School, The University of Adelaide, Adelaide, SA 5006, Australia; joshua.mcdonough@unisa.edu.au; 3Mental Health and Suicide Prevention Research and Education Group, Clinical and Health Sciences, University of South Australia, Adelaide, SA 5000, Australia; 4Department of Paediatric Infection and Immunity, Monash Health, Melbourne, VIC 3168, Australia; margaret.angliss@monashhealth.org (M.A.); jim.buttery@monash.edu (J.B.); 5Department of Paediatrics, Monash University, Melbourne, VIC 3168, Australia; 6Infectious Disease Unit, Central Adelaide Local Health Network, Adelaide, SA 5000, Australia; david.shaw@sa.gov.au (D.S.); morgyn.warner@sa.gov.au (M.S.W.); renjy.nelson@sa.gov.au (R.N.); 7Department of Microbiology and Infectious Diseases, Flinders Medical Centre, Adelaide, SA 5042, Australia; d.gordon@flinders.edu.au; 8College of Medicine and Public Health, Flinders University, Adelaide, SA 5042, Australia; 9Faculty of Health and Medical Sciences, University of Adelaide, Adelaide, SA 5000, Australia; 10Infectious Diseases, Clinical Immunology and Allergy Division of Medicine Lyell McEwin Hospital, Adelaide, SA 5112, Australia; rory.hannah@sa.gov.au

**Keywords:** invasive meningococcal disease, health-related quality of life, adolescents and young adults

## Abstract

Background: Data on the health-related quality of life (HRQoL) for invasive meningococcal disease (IMD) survivors, particularly among adolescents and young adults (AYAs), are limited. This study aimed to investigate the in-depth experiences and impacts of IMD on AYAs. Methods: Participants were recruited from two Australian states, Victoria and South Australia. We conducted qualitative, semi-structured interviews with 30 patients diagnosed with IMD between 2016 and 2021. The interview transcripts were analyzed thematically. Results: Of the participants, 53% were aged 15–19 years old, and 47% were aged 20–24. The majority (70%) were female. Seven themes relating to the participants’ experience of IMD were identified: (1) underestimation of the initial symptoms and then rapid escalation of symptoms; (2) reliance on social support for emergency care access; (3) the symptoms prompting seeking medical care varied, with some key symptoms missed; (4) challenges in early medical diagnosis; (5) traumatic and life-changing experience; (6) a lingering impact on HRQoL; and (7) gaps in the continuity of care post-discharge. Conclusion: The themes raised by AYA IMD survivors identify multiple areas that can be addressed during their acute illness and recovery. Increasing awareness of meningococcal symptoms for AYAs may help reduce the time between the first symptoms and the first antibiotic dose, although this remains a challenging area for improvement. After the acute illness, conducting HRQoL assessments and providing multidisciplinary support will assist those who require more intensive and ongoing assistance during their recovery.

## 1. Introduction

Invasive meningococcal disease (IMD) is one of the most common causes of death due to an infectious disease in developed countries, such as Australia [1]. IMD manifests as septicemia and/or meningitis [2], and survivors can often experience permanent sequelae beyond their acute infection [3,4]. Disease sequelae can be present in both the short- and long-term recovery periods, affecting survivors’ health-related quality of life (HRQoL) [5,6].

While IMD affects all age groups, its incidence in most countries, including Australia, has a bimodal distribution, peaking in children (0–4 years) and then having a smaller peak in adolescents and young adults (AYAs) (15–25 years) [7,8,9]. IMD in AYAs is an important consideration because this is a time of increased independence, as well as the completion of secondary and tertiary education and training for entering the workforce. Additionally, this is a period of significant neurological development, with the maturation of the brain structure and neurochemical pathways [10]. The pathology of IMD can interrupt this development, causing ongoing neurological issues. Previous research has shown that AYA survivors of IMD have a lower academic performance and higher rates of mental ill health compared with controls [5,11].

Early diagnosis of meningococcal disease is difficult because the initial symptoms, such as fever, headache, muscle aches, and vomiting, are similar to common viral infections [12]. The triad of the classical features of meningitis, namely fever, neck stiffness, and altered mental state, is not always all present, resulting in health professionals missing bacterial meningitis cases [13]. Even when they are all present, appropriate and timely antibiotic therapy can be delayed [14]. Another classical sign of meningococcal disease is a hemorrhagic, non-blanching rash. However, this rash often appears later in the acute illness and not in all presentations [15]. In a UK study of adolescents aged 15–16 years with IMD, approximately 66% developed a rash, on average about 19 h after developing their first symptoms [16]. The progression of the disease is usually rapid, resulting in hospitalization and often intensive care admission. Among adolescents aged 16 years, the case fatality rate is estimated at 10.4%, increasing to 15.0% in young adults aged 28 years [17]. Reducing the time from healthcare presentation to antibiotic therapy reduces mortality for bacterial meningitis and sepsis [18,19].

IMD is a largely vaccine-preventable disease, with multi-component vaccines available for the A, C, W, and Y strains (MenACWY) and the B strain (MenB). However, few countries have publicly funded meningococcal B vaccine programs for adolescents due to unknown or unfavorable cost-effectiveness analyses, hindered by a lack of contemporary data on the disease burden [20]. One study [21] has explored the psychosocial impacts experienced by young people with IMD, but there remains a lack of published qualitative data.

HRQoL is a multidimensional concept that can be described as an individual’s personal health status [22]. Despite lacking a universal definition, most descriptions suggest that HRQoL is an individual’s subjective view of their own health status and includes the dimensions of physical, emotional, and social functioning [22]. Despite being a subjective concept, HRQoL can be measured quantitatively using generic HRQoL instruments. Another option is to develop or adapt instruments psychometrically tested to be specific to the disease of interest [23]. Tools developed for or adapted specifically to conditions such as IMD have the potential to provide unique information that is not captured by more generic instruments. However, a Delphi process that included clinicians and patient representatives revealed a preference for a generic HRQoL tool over an adapted or newly developed tool for assessing the HRQoL of survivors of IMD [23]. Currently, there is no disease-specific instrument for IMD. In the absence of a disease-specific instrument, HRQoL within specific populations can be assessed with either generic HRQoL instruments (e.g., the short-form 36 [24]) or through qualitative methods.

This qualitative study aims to explore the HRQoL and lived experience of adolescents following IMD, spanning from their initial symptoms to long-term recovery.

## 2. Materials and Methods

### 2.1. Study Design

We conducted semi-structured qualitative interviews with 17–25-year-olds diagnosed with IMD and enrolled in the Long-term Impact of Invasive Meningococcal Disease in Australian Adolescents and Young Adults (AMEND) study (NCT03798574) [25]. Interviews were conducted with IMD survivors in South Australia and Victoria by the research nurses M.M. and M.A. and the research scientist L.S. The participants were unknown to the interviewers prior to their study appointments. The study’s aims were described to the participants prior to them giving consent on written participant information sheets and in verbal conversation.

### 2.2. Recruitment

This research was approved by the Human Research Ethics Committee of the Women’s and Children’s Health Network (HREC/14/WCHN/024) and Monash Health (16093A). Eligible participants were prospectively and retrospectively identified through hospital records and by admitting physicians. All eligible retrospective participants were sent a letter inviting them to participate in the study. To be eligible, participants were required to meet the following criteria: confirmed IMD diagnosis by culture or polymerase chain reaction of *Neisseria meningitidis* in the blood or cerebrospinal fluid and being aged between 15 and 25 years at the time of IMD hospitalization. Potential participants were excluded if they were not fluent in English or had a known pre-existing intellectual disability or intracranial pathology.

Eligible participants were sent a letter inviting them to participate in the AMEND study or were recruited prior to discharge from the hospital. Consent was obtained prior to the interview. The participants received reimbursement for the time they spent completing the study process, including the semi-structured interviews. Assessments were performed between 2 and 10 years post-IMD diagnosis.

### 2.3. Data Collection

Face-to-face interviews were conducted individually at Women’s and Children’s Hospital in Adelaide and Monash Health in Melbourne. Each interview was audiotaped and transcribed with only the participant and the interviewer present. Transcripts were not returned to the participants for comments or corrections. The interviews, facilitated by an interview schedule (see Appendix A) with open-ended questions, encouraged participant-directed responses. The questionnaire was developed by the study investigators and reviewed by an external qualitative research expert. Following the initial round of interviews, minor adjustments were made to refine the questions. The duration of the interviews ranged from 10 to 40 min. The participants were asked to describe how their experience with IMD had affected their HRQoL from the time they were hospitalized to the date of the interview. When needed, probing questions regarding the physical, emotional, and social aspects of HRQoL were used to elicit responses from participants. Data were collected until saturation was reached [26]. A case note review was conducted for the baseline characteristics. The Index of Relative Socioeconomic Disadvantage (IRSD) was used to classify the participants’ socioeconomic statuses based on postcode [27], and the Charlson Comorbidity Index was used for pre-existing comorbidities [28]. Participants were assigned a diagnosis of septicemia, meningitis, or mixed meningitis and septicemia by a pediatrician external to the study.

### 2.4. Analysis

The transcripts were thematically analyzed [29] using NVivo 12 software (QSR International, Chadstone, Victoria, Australia) following a critical realist paradigm. Critical realism recognizes that the world functions within a complex, multidimensional system, and therefore the phenomena that patients experience are likely to be informed by their generation of meaning and the broader structures within which patients exist [30]. The analysis was led by one author (J.M.), with each step audited by another (M.M.) to ensure consistency in the analysis. The transcripts were read several times to gain familiarity with individual accounts, with preliminary notes on salient topics. The coding followed an inductive approach, wherein the codes were driven by the data rather than a predetermined framework. Thus, the coding of earlier transcripts informed subsequent transcripts, and earlier transcripts were revisited when new themes were identified, ensuring coding consistency. Once each transcript had been analyzed, the codes were merged to form themes. Regular conversations within the research team ensured that the identified themes were captured and supported by the participant views and all relevant themes were included in the analysis. The quotes presented were selected as the most concise and comprehensive illustrations of the themes and labelled with participant numbers.

### 2.5. Role of the Funding Source

Pfizer funded the AMEND study to evaluate the long-term impact of IMD on the general intellectual functioning and quality of life of Australian AYAs, as described in the ClinicalTrials.gov clinical registry [25]. Pfizer reviewed the study protocol, but this funder had no role in the study design, data collection, or analysis and had no involvement in the development of this publication.

## 3. Results

Thirty young people participated in the interviews (see Table 1 for the participant characteristics). Prior to IMD, the majority of the participants had no pre-existing comorbidities. More than half of the participants (60%) were classified as having meningitis, with 40% having sepsis without meningitis. Over half of the participants were from the least socioeconomically disadvantaged residential areas. All had serogroup B IMD.

Our thematic analysis identified seven themes from the onset of meningococcal disease to recovery.

A.Potential barriers to timely first dose of antibiotics
(1)Underestimation of initial symptoms and then rapid progression of symptoms(2)Reliance on social support for emergency care access(3)Symptoms prompting seeking medical care varied, with some key symptoms missed(4)Challenges in early medical diagnosis
B.The life-changing impact of meningococcal disease
(5)Traumatic and life-changing experiences of IMD
C.Ongoing HRQoL issues and impacts of IMD
(6)IMD’s lingering impact on health-related quality of life(7)Gaps in the continuity of care post-discharge for patients and carers


These themes are further described below.

A.Potential barriers to timely first dose of antibiotics, Figure 1 and Table 2.B.The life-changing impact of meningococcal disease (Table 3)C.Ongoing HRQoL issues and impact of IMD (Table 4)

## 4. Discussion

This is the first qualitative study to examine HRQoL issues and lived experience in AYAs following IMD. This qualitative research provides an in-depth understanding of AYAs’ experience and the impact of IMD on their HRQoL. The themes from this study identified the life-changing experience of IMD and the lingering impact it had on both the participants in this study and their families and carers, as well as the potential barriers to timely diagnosis and treatment, which represent targets for intervention.

Reducing the time to starting antibiotic therapy reduces mortality for bacterial meningitis [18,31]. The thematic analysis identified that the period from meningococcal symptoms to receiving antibiotics remains an area that could be improved. Most research focuses on the “door-to-needle” time, where the clock starts from hospital presentation to the first appropriate dose of antibiotics [18]. A Croatian study investigating the time from the onset of symptoms to the first appropriate dose of antibiotics found that only 20% of its patients with bacterial meningitis received the appropriate treatment during the first 24 h of the disease onset. They could not investigate the reasons behind these delays [14]. This analysis suggests that some of the reasons behind the delays in healthcare presentations for AYAs include underestimation of the seriousness of their initial symptoms and then rapid deterioration, which often leaves AYAs reliant on help from others to access medical care.

In some cases, the classical signs of meningococcal disease and meningitis were either missed or not considered serious by the participants, including neck stiffness and hemorrhagic rash. The Defeating Meningitis by 2030 global road map, developed by a World Health Organization task force, includes the strategic goal “ensure awareness, among all populations, of the symptoms, signs and consequences of meningitis so that they seek appropriate health care” [32]. One of the key activities in the strategic plan calls for understanding the factors that facilitate or act as barriers to seeking healthcare. This analysis suggests that for AYAs, knowledge of distinct symptoms remains a barrier to seeking healthcare, as does reliance on others, during a time when many are becoming increasingly independent and less reliant on their families. Enhancing public awareness about key meningococcal signs, such as a stiff neck and rashes, may improve the outcomes for AYAs with IMD [33]. Future research should seek to quantitatively assess these dynamics in a larger cohort, exploring the relationship between diagnostic delays, symptom presentation and progression, and patient outcomes.

Delays in effective treatment for more than two hours after presenting to a hospital are associated with more than double the odds of death in patients with community-acquired bacterial meningitis [18]. This qualitative analysis did not investigate the reasons behind the delays in receiving an appropriate antibiotic following initial medical presentation. However, several participants had significant delays in receiving antibiotics. The causes of delays are most commonly considered to be an atypical presentation and waiting for investigations, such as lumbar puncture [31]. Some participants may not have had signs of meningitis, a hemorrhagic rash, or septicemia at their initial presentation. In two cases, neck stiffness and rashes were present but not conveyed to the treating doctor by the AYAs, suggesting some of the questions needed for an accurate assessment of IMD were not asked, nor was a thorough physical assessment conducted. A Dutch study demonstrates that the classic triad of fever, neck stiffness, and altered mental status is commonly fully not present in adults with community-acquired bacterial meningitis. They reported that 95% of their cohort had at least two of the four symptoms of headache, fever, neck stiffness, or altered mental status [13]. In this study, most of the delays were reported after presenting to a primary healthcare setting. Quality improvement processes for primary and acute care should be initiated or continued to improve the management of meningitis and acute sepsis cases [34].

Almost all of the participants in this study were hospitalized prior to a nationally funded MenACWY adolescent school vaccination program that has run since 2019. In South Australia, a state-funded MenB school immunization program for South Australian adolescents was launched in February 2019 [35], with Queensland commencing a program in 2024 [36]. Maintaining high vaccination coverage against IMD, tailored to the local epidemiology and burden, is considered the number one strategy for improving the outcomes for risk groups [32].

The participants described the HRQoL issues they experienced during their hospitalization as well as beyond their acute infection, including the traumatic elements of the disease, described as often having inadequate follow-up. The disease sequelae identified by the participants following hospitalization in our study spanned across the physical, emotional, and social aspects of HRQoL. They included pain, fatigue, difficulty concentrating, persistent headaches, a lack of motivation, poor mood, changes to hobbies, the inability to work, and changes to relationships with family and friends, among others. These experiences are consistent with the published literature and our clinical understanding of IMD sequelae and adds to our understanding of the HRQoL impacts of IMD [4]. Further qualitative and quantitative research exploring HRQoL issues in AYAs hospitalized for IMD could be conducted to help generate further understanding of the disease’s effects and promote a more holistic approach to research, as well as disease prevention and management.

Experiencing IMD during adolescence can have a particularly negative impact on a person’s HRQoL. Pain, fatigue, depression, and low motivation limit people’s ability to engage with work, education, and training, as well as limiting how people socialize. If unaddressed, these can present as issues for people long after hospitalization and potentially negatively impact their HRQoL for the remainder of their lives. Therefore, there should be some consideration of how IMD cases are managed after they have been discharged from hospital. The participants in this study identified the lack of follow-up and information regarding their recovery as an area of unmet need. Interventions aiming to empower people post-hospitalization and manage their recovery may help reduce the distress caused by the fear of the unknown and improve HRQoL [37]. All AYAs admitted with IMD should be given an individual plan for future care, regardless of their condition at discharge. Detailed information should be provided to their GP, who can assist with support and recovery and play a key role in assisting their return to study or work after the acute phase of their illness.

There is no disease-specific HRQoL instrument for AYAs hospitalized with IMD. Such an instrument could be used within hospitals and general practice to screen for those with significant deficits in their HRQoL and identify patients requiring ongoing care. Additionally, an IMD-specific HRQoL instrument could benefit future research aiming to understand better the impact of HRQoL on AYAs who become hospitalized due to IMD. Barriers to using HRQoL instruments within clinical practice have been documented at the patient, clinician, and service levels due to time constraints, unfamiliarity with patient-reported outcome measures, and a lack of infrastructure to support ongoing HRQoL measurements [38]. Despite these barriers, there is still an opportunity for future research to develop and test disease-specific instruments for IMD to aid future clinical care and research efforts.

Negative HRQoL impacts also need to be considered for the parents and caregivers of adolescents diagnosed with IMD. It is well documented that carers of people with significant illnesses have unmet support needs, which has also been identified in carers of people hospitalized with IMD [39]. The participants in this study described their caregivers’ ongoing issues as a response to their hospitalization. Caring for their child as they experience potential morbidity and mortality can impact a carer’s social and emotional functioning. A Delphi study highlighted the importance of considering not just the QoL losses experienced directly by patients but also the effects on their families, especially for 6–18-year-olds, where the impact on parents and siblings is particularly significant [40]. In this study, we only have the perspectives of the IMD cases. However, many were concerned about the effects of their illness on their immediate family. There may be a need for post-IMD infection interventions for caregivers to manage these feelings and improve their emotional functioning, as carried out for carers of patients with other diseases [41,42]. Furthermore, the clinical management of patients with IMD could consider shifting towards a multidisciplinary approach, where patients are treated during their acute infection but are also provided the opportunity for follow-up services that address the concerns of patients and carers beyond their hospitalization. This multidisciplinary approach has proven effective in other conditions, such as cancer [43], and may help center HRQoL in treating patients with IMD.

IMD can be a distressing experience for AYAs and their caregivers, with ongoing HRQoL impacts. Despite being a vaccine-preventable disease, many countries do not include a publicly funded meningococcal vaccine program for adolescents or have limited access to meningococcal vaccines [25]. However, there is some evidence supported by the findings of our study to indicate that the longer-term disease burden of IMD has been underestimated [25]; thus, a detailed and coordinated collection of disease burden data is warranted to better inform public vaccination policy for meningococcal disease.

### Limitations

Our research demonstrates that patients who are diagnosed with IMD may experience long-term HRQoL implications as a result of the disease. Due to the nature of qualitative research, our sample’s experiences are unlikely to broadly represent patients with IMD, particularly as the participants were recruited from two Australian states: South Australia and Victoria. To further understand the burden of IMD, our data should be combined with quantifiable clinical and patient-reported data from across Australia to understand better the societal and economic impact of IMD, which better informs disease prevention and management. The research participants were primarily from the least disadvantaged decile in terms of socioeconomic disadvantage. In countries similar to Australia, this is a common problem, with socially disadvantaged groups often under-represented in research [44]. Our results may not fully capture the intersection between IMD and socioeconomic disadvantage. Further research targeting more disadvantaged socioeconomic groups would be needed to investigate whether this cohort has unique experiences requiring specialist intervention and support.

## 5. Conclusions

Seven themes relating to the participants’ experience of IMD were identified in this analysis. They identified some barriers to seeking timely healthcare access. They also identified the profound impact IMD had on the adolescents and their families, including a lingering impact on their quality of life for some. Post-discharge care was highlighted as an area that could be enhanced for AYAs following discharge.

A multifaceted approach should be considered to improve the experience and outcomes for AYAs with IMD. These approaches should include (a) improving early detection through increasing awareness of the key IMD symptoms for AYAs and continuing education and conducting quality improvement processes for primary and acute care to ensure the timely management of meningitis and sepsis, (b) developing follow-up care plans based on HRQoL screening of AYAs at discharge, (c) ensuring these plans include the input of members of a multidisciplinary team, and (d) conducting a questionnaire or follow-up with the parents and caregivers of AYAs to gain additional insights into the family experience with meningococcal disease, which could enhance care.

## Figures and Tables

**Figure 1 healthcare-12-01075-f001:**
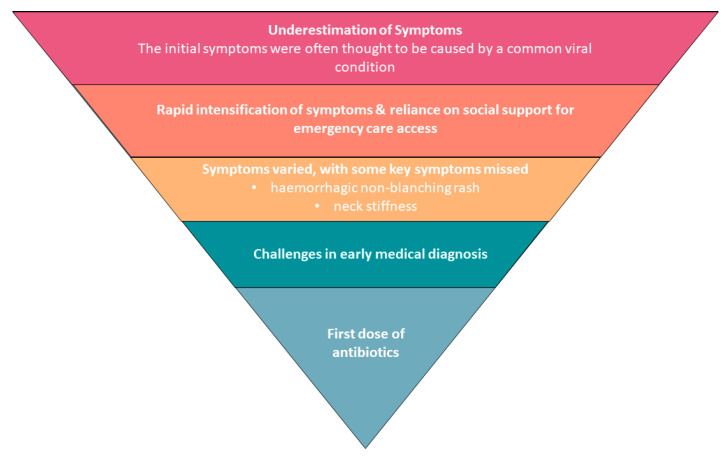
Barriers to timely first dose of antibiotics.

**Table 1 healthcare-12-01075-t001:** Patient characteristics (N = 30).

Participant Characteristics		N (%)
Sex		
	Female	21 (70%)
	Male	9 (30%)
Location		
	South Australia	21 (70%)
	Victoria	9 (30%)
Age at admission		
	15–19	16 (53%)
	20–24	14 (47%)
IRSD (deciles) *		
	1–2 (most disadvantaged)	2 (7%)
	3–4	6 (20%)
	5–6	1 (3%)
	7–8	4 (13%)
	9–10 (least disadvantaged)	17 (57%)
Charlson Comorbidity Index score		
	0 (No pre-existing comorbidity)	28 (93%)
	1	2 (7%)
	>2	0 (0%)
Disease type		
	Septicemia	12 (40%)
	Meningitis	2 (7%)
	Mixed meningitis and septicemia	16 (53%)
Meningococci serogroup		
	B	30 (100%)

* Index of Relative Socioeconomic Disadvantage.

**Table 2 healthcare-12-01075-t002:** Themes and IMD survivor responses identifying potential barriers to a timely first dose of antibiotics.

Theme	Description	Responses
1. Underestimation of initial symptoms and then rapid progression of symptoms	The initial symptoms were often thought to be caused by a common viral condition.	I kind of just felt like I had flu-like symptoms—P11 I just remember waking up one morning feeling quite unwell. Um, it was just cold symptoms, just viral symptoms. I had like a runny nose—P27
	Many participants described a rapid intensification of their symptoms, indicating to them, or others, that their condition was more serious than initially thought.	I drove home …, which is about 20 min, and I was crying the whole time because I knew something wasn’t right, and by the time I got home, I couldn’t get the key in the backdoor—P15 It was a school day. I started to get a really splitting headache, and I just thought I was getting a migraine—P24
2. Reliance on social support for emergency care access	Many participants described receiving assistance from family or friends to access medical care. This help was often required due to their rapid deterioration, which rendered them unable to help themselves. The participants highlighted their family, friends, and caregivers’ vital roles in identifying their condition’s seriousness and initiating and facilitating access to emergency medical services.	So, normally, when I’m sick and I don’t go to work and everything, mum just leaves me, shuts my door, doesn’t even come in, but for some reason, she came in my room that day. I was covered in vomit, my eyes were rolling back, and she rang the ambulance—P11 I was like no, no I’m fine, it’s just the flu. Eventually, he [dad] forced me to go to the GP, who then was like, no, you need to go to hospital—P1 Mum stayed up. I think she fell asleep on the couch, but something woke her up around 3.00 o’clock, and she came in, and I was collapsed halfway across my bedroom and would start fitting if she touched me. I don’t really remember any of that. Woke up in ICU—P16 That night, I was taken to the hospital. I couldn’t get to the car, I had to be carried to the car—P17 I remember my mum came in to check on me to see what I was doing. And she told me to stop getting ready because I wasn’t well. Mum called the GP and the GP said it didn’t sound very good, to go straight to the hospital—P21 Mum dragged me to the doctor’s appointment. She was like, no, you’re like actually sick. We need to go. Um, and when I was sitting in the waiting room at the GP, my neck started to get really stiff—P4 I thought I had the flu, went to sleep. Then, apparently, mum came and checked on me about midday and sat me up, and basically I was unresponsive to her. I just wasn’t there—P8
3. Symptoms prompting seeking medical care varied, with some key symptoms missed	The symptoms prompting seeking medical care varied and ranged from confusion or loss of consciousness to severe gastrointestinal symptoms, rash, headaches, neck stiffness, seizures, and joint pain, many in combination with each other. In some instances, a hemorrhagic non-blanching rash, a distinct symptom associated with meningococcal disease, or neck stiffness, a key sign of meningitis, did not prompt individuals to seek urgent medical assistance.	I remember seeing it [rash] on my legs during the afternoon, but I didn’t really think anything of it—P16 I got out of the shower and I noticed I was a bit spotty, but I wasn’t in the right sense of mind…. I didn’t take any notice of it because I wasn’t in the right mind to self-evaluate if it was that serious or not—P14 [I had] fever and body ache all over and my neck was a bit stiff as well. [I] still thought it was the flu—P1 I was at home by myself and …..my neck was really, really stiff [when I woke up]….. and I thought I had chickenpox….. because I had a rash all over my body. I [stayed] in bed all day—P26
4. Challenges in early medical diagnosis	Some participants had delays in treatment despite presenting to a healthcare provider. Ten participants reported having more than one presentation to a healthcare setting before they received an accurate diagnosis and appropriate treatment. These were primarily presentations to a general practitioner, and to a lesser extent, in acute care settings.	I told [my General Practitioner] about the fever and everything—so yeah, he just gave me Maxolon and sent me home—P30 [The] doctor saw me and she said oh, it looks like you’re coming down with the flu, just go home, take some Nurofen, you’ll be fine—P25 And then by the time the home doctor comes, I can’t really use my legs anymore. So, I’m kind of like crawling or like, you know, holding onto things, drag myself through the door to let him in. He said that he thought I might have a bladder infection. So to wait until tomorrow and get some antibiotics. So, [he] gave me a prescription and left—P6 I went to the doctors, and I just explained to him that I was vomiting and... he gave me some medicine [Maxalon]. I’m sure if I probably told him about the rash, then that might have prompted something—P30 I was in the waiting room with everyone else for, like, I don’t know how long. Um, and then I was, like, in a lot of pain by then, ‘cause of, like, the headache, and they gave me a Panadol, but then I was, like, screaming, kind of, ‘cause it was very painful—P2

**Table 3 healthcare-12-01075-t003:** Themes and IMD survivor responses to the impact of meningococcal disease.

Theme	Subthemes	Description	Responses
5. Traumatic and life-changing experience of IMD	Difficulty coming to terms with their experience	Due to the severity of their condition and treatment, many participants had a scant recollection of their intensive care admission when they were critically unwell. However, they were often confronted at a later stage with how close they came to death. Some are still struggling to come to terms with their experience.	Just [my family’s] story about having [me] 10 percent to live from the doctors. That was probably more scary hearing that—P14 I think I was more frightened after it all happened. I’d got home, and I went, oh, I nearly died—P6 I’d say it put me in a little bit of a mental merry-go-round because it was such a traumatic experience to have to go through; it’s something I still don’t quite understand—P7 I’m a very young person, and in an ICU unit, most people die….It was very traumatic, probably much more so than I’d ever let on—P9
	Adverse impact on caregivers	As well as being traumatic for the participants, having a child experience IMD had traumatic effects on the caregivers as well. The participants described the ongoing issues their parents experienced as they recovered from their infection.	My mum’s probably like the worst affected by it. She can’t really even talk about it. She just starts crying—P18 I think it took a while for them to get over the whole near-death experience for my parents and the fact that they almost lost their child and it affected my dad, my stepmum, and my step siblings—P28

**Table 4 healthcare-12-01075-t004:** Themes and IMD survivor responses—ongoing HRQoL issues and impact of IMD.

Theme	Subthemes	Description	Responses
6. IMD’s lingering impact on health-related quality of life	Lingering psychological impacts	Beyond the impacts of the initial hospitalization for IMD, the participants described the ongoing HRQoL challenges they had or were continuing to experience. Many survivors reported psychological impacts such as depression and anxiety that persisted up to and beyond one year post-hospitalization.	I had a massive, panic attack. I was just like, so uncontrollably upset and it was hard to describe—P18 Physically, [I’m better]; as for mentally, I had depression for years following—P29 I’ve had a few anxiety problems in the last year—P28
	Lingering effects on memory and concentration	Survivors also reported cognitive challenges, including issues with memory loss and concentration difficulties.	[my] memory, it’s not as good as it used to be—P15 No one was able to give me a good indication of whether I would have any mental function or not. I had no memory and very little concentration.—P9 I still struggle with just concentrating—P27
	Lingering physical effects	Chronic headaches, pain, and fatigue were the next most commonly reported sequelae by the survivors. Other reported sequelae included weakened immune systems, tinnitus, lower fitness levels, poor sleep, skin rashes, photosensitivity, weight loss, and altered temperature regulation.	I would have at least one severe migraine a day. But the entire time that I didn’t have a migraine, my headache was almost at migraine level—P11 Like I’m tired, but I can get through the day. Headaches are a real issue—P16 I was getting daily migraines, it just never stopped. So, I had to go from full-time to part-time at work. I had to have a lot of adjustments done to my work because I was in [an] office. But that really affected my mental health. Because it would have stopped my career—P6 I was absolutely exhausted for like a year. There’s still like a limit on how much I can do in the day without just crashing for a week afterwards—P4 Before meningococcal, I hardly ever got sick. Then all of a sudden [after IMD], everything hit me. I got sick every day. I was falling down ill with something—P23
	Impact on education and employment	Lingering psychological, cognitive, and physical impacts disrupted the education and professional lives of many of the survivors. These impacts range from minor disruptions to significant hardship in returning to full-time work or study.	I didn’t work for two months just because just having conversations with people was exhausting. I’d go for coffee with one friend, but even being in a room and everyone else talking, even though they weren’t talking to me, and then trying to focus on what she was saying or trying to was just like lots for me—P18 I was in Year 12 that year. So, I did one subject that year, and then I finished it last year—P24 [I] lose concentration so easily and it’s so hard. I think that’s kind of scared me. I want to go back to studying, but I’m just scared—P28 I was supposed to only do three years [at Uni], and I ended up doing four, and I had to repeat a lot of those subjects. Uni was a lot harder [after IMD]—P3 Going back to university was a real struggle—P9 So, I went back and did half days at work. I was useless. Later, when I think back on it, I should not have—P20 I wasn’t able to do full-time work, which was really hard on me financially as well—P6
7. Gaps in the continuity of care post-discharge for patients and carers		Participants described the time after being discharged from the hospital as a time of unmet health needs. Most of the participants reported high-quality care whilst in hospital despite the effects of IMD. However, some felt that the level of care and follow-up they received after being discharged was not proportionate to the severity of the effects of IMD and the ongoing issues the participants experienced.	While you were in hospital, everyone was really good … but I felt like it kind of all just ended really quickly after I came out of hospital. There was no follow-on.—P28 I had a follow-up two weeks later, and then that’s the last kind of thing they ever followed me up about—P11 Mum really struggled to get me a follow-up appointment at the hospital—P5 They didn’t say anything. I just had the two check-ups, and they tested how swollen my brain was still, and that’s—that’s it—P3
		As well as experiencing ongoing adverse effects from IMD, the participants cited the lack of follow-up as compounding their distress due to fear of the unknown. With limited follow-up and resources, some participants were unable to reconcile their experiences, as they had no frame of reference for what was expected and whether they would fully recover.	I just didn’t know what to expect, if what I was experiencing was normal—P27 I don’t think [I got enough information], to be honest … when I went home, I was looking [IMD] up and I was like ‘oh!’—P25

## Data Availability

The data presented in this study may be made available on request by the corresponding author. Individual participant data (IPD) will be provided on a case-by-case basis at the discretion of the study investigators and the WCHN HREC to maintain confidentiality. Access to IPD will be granted solely for the purposes outlined in the approved proposal.

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
