# Peer review of "Exploring the Health-Related Quality of Life and the Lived Experience of Adolescents Following Invasive Meningococcal Disease"

_healthcare, 2024, doi:10.3390/healthcare12111075_

Round 1
Reviewer 1 Report
Comments and Suggestions for Authors
The study describes a study concerning the quality of life in patients after meningococcal invasive disease in Australia. The introduction contains all the necessary information regarding meningococcal disease and the aim of the study. Methods are clearly described. The study was prepared well. However, I consider the part presenting the result insufficient.
The authors cite selected patients' opinions, not the full spectrum of possible responses. I doubt whether the presented opinions support the author's point of view since I am unsure if there are any others. I would suggest stratifying all received answers and presenting them as groups of responses. In my opinion, the conclusions would be better supported that way.
The discussion is written well.
Author Response
Dear Reviewer 1,
Thank you very much for your time and helpful feedback. We have addressed each suggestion below.
Comments: The authors cite selected patients' opinions, not the full spectrum of possible responses. I doubt whether the presented opinions support the author's point of view since I am unsure if there are any others. I would suggest stratifying all received answers and presenting them as groups of responses. In my opinion, the conclusions would be better supported that way. The discussion is written well.
Response: We have revised our results section to include a more detailed stratification of the responses. Sections B and C now categorise responses into distinct groups, including further quotes. We have changed the overall heading for Section A to define it more clearly as addressing potential barriers to the timely administration of the first dose of antibiotics. Highlighting areas for improvement in clinical practice.
Reviewer 2 Report
Comments and Suggestions for Authors
Dear authors;
Thanks a lot for your work and efforts done in this paper. The topic is very interesting and a hot topic and highlights the drawbacks of delayed diagnosis. I have some comments that needs to be addressed:
1- In the abstract line 29: years should be added
2- Introduction is very good and informative
3- I have some concerns about the questionnaire: was it a validated one or how did the author generate it.
4- In the results: the presentation of the results is really nice, but i have some concerns about patients' testimonial: I either suggest collecting them in one table and dividing in not them 1,2,3 etc. or adding a frame around it after each section.
5- I need some figures to be added that would be informative and helpful for readers as on the duration of diagnostic lag, outcome in an objective way, most common presenting symptoms, duration between presentation and symptoms escalation
Again many thanks for this article
Author Response
Dear Reviewer 2,
Thank you very much for your time and helpful feedback. We have addressed each suggestion below.
1- In the abstract line 29: years should be added
Response: added ‘years’
2- Introduction is very good and informative
3- I have some concerns about the questionnaire: was it a validated one or how did the author generate it.
Response: The questionnaire used in our study was developed by the study investigators to specifically address the research questions. We also engaged an external qualitative research expert, Professor Braunack-Mayer, who reviewed and provided critical feedback on the instrument.
Following the initial round of interviews, we conducted an assessment of the questionnaire to confirm that it was capturing the required information. Based on this assessment, minor adjustments were made to refine the questions.
The following wording has been added to the methods: Line 129 (tracked version): The questionnaire was developed by the study investigators and reviewed by an external qualitative research expert. Following the initial round of interviews, minor adjustments were made to refine the questions.
4- In the results: the presentation of the results is really nice, but i have some concerns about patients' testimonial: I either suggest collecting them in one table and dividing in not them 1,2,3 etc. or adding a frame around it after each section.
Response: We have stratified some themes into sub-themes for sections B and C and now categorise responses into distinct groups, including further quotes. (as suggested by reviewer 1), and presented the results in table form for clarity (as suggested by reviewer 3).
5- I need some figures to be added that would be informative and helpful for readers as on the duration of diagnostic lag, outcome in an objective way, most common presenting symptoms, duration between presentation and symptoms escalation.
Response: Thank you for your suggestions regarding the inclusion of additional figures to enhance the manuscript. We have updated Table 1 to include variables such as socioeconomic status, Charlson Comorbidity Index, disease type, and serogroup.
Unfortunately, we do not have the data necessary to calculate the duration of diagnostic lag or to directly link it with patient outcomes. Similarly, the qualitative nature of our study limits our ability to provide quantifiable lists of specific symptoms. However, we agree these are important. We have added the following sentence to the discussion on line 232. “Future research should seek to quantitatively assess these dynamics in a larger cohort, exploring the relationship between diagnostic delays, symptom presentation and progression, and patient outcomes.”
To address the need for information on barriers to timely antibiotic administration, we have added a new figure (referenced on line 192), which illustrates the key barriers identified in our study.
Reviewer 3 Report
Comments and Suggestions for Authors
The authors presented a study focusing on the health-related quality of life (HRQoL) of adolescents and young adults who are survivors of invasive meningococcal disease.
The findings highlight several key points , mainly the underestimation of initial symptoms followed by rapid escalation, variability in symptoms prompting medical care-seeking, challenges in early diagnosis, lingering impacts on HRQoL, and gaps in continuity of care post-discharge.
- It's important to elaborate more on the role of the funder, even if there's no obvious conflict of interest regarding the analysis of the results. It would be useful to provide details on their financial participation and in which specific areas it focused.
-The inclusion criteria appear to lack specificity. It would be necessary to clarify whether patients with sequelae, organic pathologies, other comorbidities, or long-term treatments were included in the study.
- The main issue lies in the visibility of the results, which is a common challenge in quality of life studies. It would be preferable to improve the visibility of the results to make them clearer ( tables, charts, or diagrams).
- Table 1 : more details , history, comorbidities , financial ?
- Overall, even if the study is interesting, as we could all predict of it : all patients undergoing an acute disease, will have an impacted QoL.
Comments on the Quality of English LanguageMinor editing
Author Response
Dear Reviewer 3,
Thank you very much for your time and helpful feedback. We have addressed each suggestion below.
- It's important to elaborate more on the role of the funder, even if there's no obvious conflict of interest regarding the analysis of the results. It would be useful to provide details on their financial participation and in which specific areas it focused.
Response: The funding covered the costs associated with study administration, recruitment, medical, neuropsychological, and audiology assessments of AYAs between 2 and 10 years post-acute illness from meningococcal disease.
As described on line 159, the funder did not have any role in the design of the study, the collection, analysis, or interpretation of the data, nor were they involved in the writing or decision to submit this publication.
The qualitative component of our study, which aimed to provide a deeper understanding of the survivor’s experiences, was independently designed and implemented by our research investigators.
We have added this to line 159: Pfizer funded the AMEND study to evaluate the long-term impact of IMD on the general intellectual functioning and quality of life of Australian AYAs, as described in Clini-calTrials.gov clinical registry [25].
The inclusion criteria appear to lack specificity. It would be necessary to clarify whether patients with sequelae, organic pathologies, other comorbidities, or long-term treatments were included in the study.
Response: We have added the exclusion criteria after the inclusion criteria to clarify the criteria. Anyone fitting the criteria was eligible, and invitations were sent to all IMD cases. There were no other criteria.
Added to line 111: All eligible retrospective participants were sent a letter inviting them to participate in the study.
Added to line 115: Potential participants were excluded if they were not fluent in English or had a known pre-existing (prior to IMD) intellectual disability or intracranial pathology.
The main issue lies in the visibility of the results, which is a common challenge in quality of life studies. It would be preferable to improve the visibility of the results to make them clearer ( tables, charts, or diagrams).
Response: We have added tables for the results and a diagram as suggested.
- Table 1 : more details , history, comorbidities , financial ?
Response: We have added socioeconomic status, Charlson Comorbidity Index, Disease type, and serogroup to table 1.
Round 2
Reviewer 3 Report
Comments and Suggestions for Authors
Authors made major changes in their manuscript.
It could be accepted in this form.
Comments on the Quality of English LanguageMinor editing.